# Ring Finger Protein 125 Is an Anti-Proliferative Tumor Suppressor in Hepatocellular Carcinoma

**DOI:** 10.3390/cancers14112589

**Published:** 2022-05-24

**Authors:** Takahiro Kodama, Michiko Kodama, Nancy A. Jenkins, Neal G. Copeland, Huanhuan Joyce Chen, Zhubo Wei

**Affiliations:** 1Houston Methodist Research Institute, Houston Methodist Hospital, Houston, TX 77030, USA; mkodama@gyne.med.osaka-u.ac.jp (M.K.); njenkins1@mdanderson.org (N.A.J.); ncopeland1@mdanderson.org (N.G.C.); 2Department of Gastroenterology and Hepatology, Osaka University Graduate School of Medicine, Suita, Osaka 5650871, Japan; 3Department of Obstetrics and Gynecology, Osaka University Graduate School of Medicine, Suita, Osaka 5650871, Japan; 4Department of Genetics, The University of Texas MD Anderson Cancer Center, Houston, TX 77030, USA; 5The Pritzker School of Molecular Engineering, The University of Chicago, Chicago, IL 60637, USA; joycechen@uchicago.edu; 6The Ben May Department for Cancer Research, The University of Chicago, Chicago, IL 60637, USA; 7Institute of Biosciences and Technology, Texas A&M University, Houston, TX 77030, USA

**Keywords:** hepatocellular carcinoma, driver genes, RNF125, tumor suppressor, anti-proliferation, whole-transcriptome analysis

## Abstract

**Simple Summary:**

HCC is a leading cause of cancer-related deaths worldwide. However, its tremendous inter- and intra-tumor heterogeneity has made it difficult to identify driver genes for HCC. Transposon mutagenesis is a versatile in vivo tool to identify cancer genes in tissues of interest. Herein, we analyzed transposon mutagenesis screening data in the liver and discovered a novel tumor suppressor gene, RNF125. This gene functions as a negative regulator of cell proliferation through transcriptional suppression of multiple genes important for cell proliferation and liver regeneration. This gene is frequently inactivated in human HCC and has a significant impact on patient prognosis. Our findings identify a new regulatory network of cell proliferation mediated by RNF125 and its contribution to HCC development and progression.

**Abstract:**

Hepatocellular carcinoma (HCC) is one of the deadliest cancers worldwide and the only cancer with an increasing incidence in the United States. Recent advances in sequencing technology have enabled detailed profiling of liver cancer genomes and revealed extensive inter- and intra-tumor heterogeneity, making it difficult to identify driver genes for HCC. To identify HCC driver genes, we performed transposon mutagenesis screens in a mouse HBV model of HCC and discovered many candidate cancer genes (SB/HBV-CCGs). Here, we show that one of these genes, RNF125 is a potent anti-proliferative tumor suppressor gene in HCC. RNF125 is one of nine CCGs whose expression was >3-fold downregulated in human HCC. Depletion of RNF125 in immortalized mouse liver cells led to tumor formation in transplanted mice and accelerated growth of human liver cancer cell lines, while its overexpression inhibited their growth, demonstrating the tumor-suppressive function of RNF125 in mouse and human liver. Whole-transcriptome analysis revealed that RNF125 transcriptionally suppresses multiple genes involved in cell proliferation and/or liver regeneration, including Egfr, Met, and Il6r. Blocking Egfr or Met pathway expression inhibited the increased cell proliferation observed in RNF125 knockdown cells. In HCC patients, low expression levels of RNF125 were correlated with poor prognosis demonstrating an important role for RNF125 in HCC. Collectively, our results identify RNF125 as a novel anti-proliferative tumor suppressor in HCC.

## 1. Introduction

Hepatocellular carcinoma (HCC) is the sixth most common cancer and the third leading cause of cancer-related deaths worldwide [1]. The incidence rate of HCC is continuously increasing in the U.S., and the overall 5-year survival is <12% regardless of the recent progress in diagnostic and therapeutic modalities [2,3]. To identify key molecules involved in the development and progression of HCC, next-generation sequencing (NGS) technology has been used to profile HCC genomes by both the International Cancer Genome Consortium (ICGC) [4] and The Cancer Genome Atlas (TCGA) [5]. Over 500 cases of HCC were sequenced, and these data are publicly available. This large collection of data has enabled us to profile the liver cancer genome and elucidate previously unexplored pathways related to HCC, such as chromatin remodeling and oxidative stress pathways [6]. However, comprehensive genomic analysis showed that only four genes, TERT, CTNNB1, TP53 and ARID2, have ≥5% frequencies of non-synonymous mutations, and numerous infrequent mutations (<5%) were identified in >10,000 genes [7]. This indicates tremendous inter- and intra-tumor heterogeneity in HCC. In addition to non-synonymous mutations, cancer genes can also be dysregulated by epigenetic alterations and copy number changes, single nucleotide polymorphisms (SNP) and alterations in functional elements identified by the Encyclopedia of DNA Elements (ENCODE) project [8]. This complex gene regulatory network makes it difficult to identify true drivers of HCC. To overcome these limitations, we performed transposon mutagenesis screens in an HBV mouse model of HCC and discovered many HCC candidate cancer genes (CCGs) [9]. Many of these genes are mutated or dysregulated in human HCC, indicating the relevance of this mouse model to the human disease. CCGs were enriched in major cancer signaling pathways, but surprisingly, genes related to metabolic pathways were also highly enriched, demonstrating the importance of disrupted metabolism in HCC development.

In this study, we used a comparative genomics approach to show that one of these CCGs, ring finger protein 125 (RNF125), is an anti-proliferative tumor suppressor in HCC. We also show that RNF125 is a transcriptional repressor of multiple genes involved in cell proliferation and/or liver regeneration, including Egfr, Met, and Il6r, and its reduced expression is correlated with poor prognosis in HCC patients.

## 2. Materials and Methods

### 2.1. Cell Lines and Reagents

HepG2, Hep3B, SNU-398 and PLC/PRF/5 human liver cancer cell lines were purchased from the American Type Culture Collection (ATCC). HEK293T cells were purchased from Thermo Fisher Scientific. Immortalized liver progenitor cells that were p53-null and overexpressed Myc, were kindly provided by S. Lowe (Memorial Sloan-Kettering Cancer Center) [10]. These cell lines were cultured in DMEM or RPMI-1640 medium supplemented with 10% FBS and penicillin-streptomycin. All cells were confirmed to be free from pathogens and mycoplasma. Erlotinib and Tivantinib were purchased from Selleckchem and dissolved in DMSO.

### 2.2. Lentiviral Production and Transduction

Lentiviral vectors containing GIPZ shRNA targeting mouse RNF125 (V2LMM_61827, V2LMM_72504) or NC shRNA were purchased from Thermo Fisher Scientific. To produce lentivirus containing GIPZ shRNA, Trans-Lentiviral shRNA packaging kit (Thermo Fisher Scientific) was used according to the manufacture’s instruction. Viral titer was determined by qPCR using qPCR Lentivirus Titration Kit (Abm) according to the manufacture’s instruction. Lentiviral particles containing GIPZ shRNA targeting human RNF125 (V2LHS_219542, V3LHS_303932) or NC shRNA were purchased from Thermo Fisher Scientific. Lentiviral particles containing the human RNF125 expression vector or NC vector were purchased from Abm and GeneCopoeia. Cells were plated at a density of 1.0 × 10^5^ cells per 6-well plate one day before the infection. The next day, medium was changed into serum-free medium with 8 μg/mL of polybrene (Millipore) and transduced with lentivirus at MOI of 1.0. The following day, cells were supplemented with complete medium including 1.0–1.5 μg/mL Puromycin (Thermo Fisher Scientific) for establishing stable cell lines with lentiviral integration. 

### 2.3. Xenograft Assay

Female athymic nude mice (Crl:Nu(NCr)-Foxn1nu) were purchased from Charles River Laboratories. 1.0–2.0 × 10^6^ cells were injected into the bilateral flanks of athymic nude mice. A total of 6 to 10 flanks were injected for each cohort. Mice were monitored for tumor growth at least twice a week and euthanized when cumulative tumor measurements reached 2.0 cm in diameter. Tumor volumes were calculated by the following formula: volume (mm^3^) = length × width × width × 0.52. Data was presented as mean ± S.E.M. All xenograft procedures were approved by IACUC at Houston Methodist Research Institute.

### 2.4. siRNA Transfection

The following siRNA oligonucleotides were purchased from Thermo Fisher Scientific; RNF125 Silencer Select siRNA (s29809, s29810, s29811) and EGFR Silencer Select validated siRNA (s564) and MET Silencer Select validated siRNA (s8701). siRNA transfection was performed using Lipofectamine RNAiMAX transfection reagent (Thermo Fisher Scientific, Waltham, MA, USA) according to the manufacture’s instruction. Knockdown efficiency was determined by qRT-PCR 2 days after transfection.

### 2.5. WST-1 Cell Proliferation Assay

The WST-1 proliferation assay was performed using the Premix WST-1 cell proliferation assay system (Takara) according to the manufacture’s protocol. Briefly, 4 × 10^3^ to 8 × 10^3^ cells were seeded into 96-well plates and cultured with medium containing 10% or 1% FBS for 1 to 5 days. WST-1 regent was added to each well and incubated for 1.5 h at 37 °C. The absorbance at 450 nm and 650 nm was measured using an Infinite 200 pro multimode reader (TECAN).

### 2.6. RT-qPCR and RNA-Seq

Total RNA was extracted from cells using the RNeasy Plus Mini kit (Qiagen, Germantown, MD, USA). Reverse transcription was carried out using SuperScript VILO Master Mix (Thermo Fisher Scientific, Waltham, MA, USA). qPCR was performed using the QuantStudio 12K Flex RT-PCR system with TaqMan Gene Expression Assay probes (Thermo Fisher Scientific, Waltham, MA, USA) for human RNF125 (Hs00215201_m1), mouse RNF125 (Mm00651227_mH), human EGFR (Hs01076090_m1) and human MET (Hs01565576_m1). For RNA-seq experiments, RNA quality was assessed using an Agilent Bioanalyzer (Agilent Technologies, Santa Clara, CA, USA). High-quality RNA was processed using an Illumina TruSeq Stranded Total RNA Sample Preparation kit according to the manufacture’s protocol. All libraries were pooled using 6-nucleotide barcoded adapters. Pooled libraries were sequenced using an Illumina Hiseq 2500. All raw reads that passed the chastity filter were analyzed and clipped using Trimmomatic (Version 0.32, http://www.usadellab.org/cms/index.php?page=trimmomatic accessed on 6 March 2016) [11]. Mapping of reads to reference sequences was performed using STAR (version 2.3.0e https://code.google.com/p/rna-star/, accessed on 6 March 2016) [12]. Raw read counts were created using HTSeq (http://www-huber.embl.de/users/anders/HTSeq/doc/overview.html, accessed on 1 May 2016) [13]. A Trimmed Mean of M-values (TMM) normalization was performed using the edgeR package (http://bioconductor.org/packages/release/bioc/html/edgeR.html, accessed on 1 June 2016) [14]. Genes with more than 10 normalized read counts in control were used for differential gene expression analysis.

### 2.7. Western Blot

Cells were lysed in RIPA buffer (Pierce) with protease and phosphatase inhibitor cocktail (Pierce) on ice. The cell lysates were cleared by centrifugation at 10,000× *g* for 15 min at 4 °C. Protein concentrations were determined using a bicinchoninic acid protein assay kit (Pierce). The protein lysates were electrophoretically separated using sodium dodecyl sulfate polyacrylamide gels and transferred onto a polyvinylidene fluoride membrane. For immunodetection, the following antibodies were used: a rabbit polyclonal antibody to EGFR, a rabbit polyclonal antibody to p-ERK, a rabbit polyclonal antibody to MET and a rabbit polyclonal antibody to β-actin (Cell Signaling Technology, Danvers, MA, USA).

### 2.8. Bioinformatic Analysis

For gene expression analysis, publicly available TCGA RNA-seq RSEM-normalized data of HCC were downloaded from the TCGA data portal (https://tcga-data.nci.nih.gov/tcga/, accessed on 6 March 2016). For miRNA analysis, 3 normalized miRNA microarray datasets (GSE6857, GSE10694 and GSE22058) were downloaded from GEO (http://www.ncbi.nlm.nih.gov/geo/, accessed on 6 March 2016). The correlation analysis between patient survival and gene expression levels was done using the published public web server, GEPIA 2 (gene expression profiling and interactive analyses version 2) (http://gepia2.cancer-pku.cn, accessed on 1 July 2021) [15]. Differential gene expression analysis between tumor and non-tumor normal tissues was performed using expression data downloaded from the TCGA database and analyzed using GraphPad Prism (http://firebrowse.org/iCoMut/?cohort=lihc on 6 March 2016). Calculated *p* values were corrected for multiple hypothesis testing by computing both the false discovery rate (FDR) and the family-wise error rate (FWER). FWER less than 0.01 was considered as statistically significant. For miRNA target prediction, the following three software packages were used; TargetScan (http://www.targetscan.org/ on 6 March 2016), microT-CDS (http://diana.imis.athena-innovation.gr/DianaTools/index.php?r=microT_CDS/index on 6 March 2016), and miRanda (http://www.microrna.org/microrna/home.do, accessed on 1 March 2016). Publicly available TCGA SNP6 copy number analysis (GISTIC2) data for HCC (http://dx.doi.org/10.7908/C1XK8DNT, accessed on 5 March 2016) was downloaded from FIREHOSE (http://gdac.broadinstitute.org/ on 6 March 2016). Mouse CISs were converted to human genes using two different annotation databases (MGI EntrezGene associations (http://www.informatics.jax.org/, accessed on 10 March 2016), and HGNC complete annotations (http://www.genenames.org/, accessed on 10 March 2016).

### 2.9. Statistics

All data were provided as mean ± S.D. unless otherwise indicated. Statistical analyses were performed using an unpaired Student’s *t*-test or one-way ANOVA using GraphPad Prism software. When the ANOVA analyses were applied, the differences in the mean values among the groups were examined by post hoc correction unless otherwise indicated. Differences with *p* < 0.05 were considered statistically significant.

## 3. Results

### 3.1. RNF125 Is a Candidate Tumor Suppressor and Significantly Downregulated in Human HCC

To identify genes driving HBV-induced HCC, we performed a Sleeping Beauty (SB) transposon mutagenesis screen in hepatocyte-specific hepatitis B surface antigen transgenic mice (HBs-Ag Tg) [9]. This mouse model has been widely used to study the effects of chronic HBV infection in HCC [16]. As revealed in this screen and other SB screens, most CCGs identified by transposon mutagenesis in solid tumors are predicted to function as tumor suppressor genes (TSGs) [9,17,18,19,20,21,22,23,24,25,26]. This may explain the significant enrichment of downregulated genes in the HCC-SB screen. For validation, we applied a more stringent statistical criteria for CCGs calling (q value < 1 × 10^−20^) in the datasets from Liver-Onc2/HBsAg and Liver-Onc2 mice and identified 599 common CCGs (576 human homologues) (Figure 1A) (Appendix A). Since these genes were selected regardless of the existence of the HBV transgene, we assumed they would be involved in more fundamental HCC processes. To identify tumor suppressor genes (TSGs) inactivated in human HCC, we focused on CCGs highly downregulated in the TCGA RNA-seq dataset of human HCC. Among 352 genes with significant downregulation of mRNA abundance in HCC (more than 3-fold in this dataset), 9 genes were common CCGs (Figure 1A). One of these genes, FOXO1 is a known TSG in the liver as well as other organs [27,28,29]. Interestingly, 7 genes were liver-specific CCGs and were not detected in transposon screens from 13 other tumor types [9,21,22,26,30,31] (and unpublished data) (Figure 1B). Many of the CCGs were involved in metabolism or encoded liver-specific proteins [32,33,34,35,36], suggesting that altered functions of these genes may be crucial for hepatocarcinogenesis. Among them, RNF125 was linked to neither HCC nor metabolism. We examined the insertion pattern of transposons in the RNF125 gene in two datasets (Liver-Onc2/HBsAg CISs and Liver-Onc2 CISs) and found that the insertions were scattered throughout the gene and in both transcriptional orientations (Figure 1C), suggesting that RNF125 is likely a tumor suppressor.

### 3.2. RNF125 Expression in Tumor Tissues Showed a Significant Negative Correlation with Survival in HCC Patients

We next examined the clinical impact of 9 CCGs in human HCC. To this end, we analyzed a publicly available TCGA dataset containing 377 human HCC samples that included patient survival data. First, we confirmed that RNF125 expression was significantly downregulated in tumor tissues compared to paired normal tissue (Figure 1D). We then analyzed the relationship between RNF125 expression and patient survival. Expression levels of RNF125 in tumor tissues showed a significant negative correlation with patient survival (Figure 1E), suggesting that RNF125 plays a crucial tumor suppressive role in human HCC and its inactivation has a significant negative impact on patient survival. Regarding the other 8 CCGs, only GHR showed significant negative association between the expression levels and patient survival (Appendix A). Since the importance of GH signaling in liver cancer development and its clinical impact have been already reported [37,38], we decided to focus on functional analysis of RNF125. We further examined the clinical relevance of RNF125 gene in human HCC using a publicly available microarray dataset containing 221 human HCC samples. Once again, the RNF125 gene was significantly downregulated in tumor tissue, and its expression levels were significantly negatively associated with patient prognosis (Appendix A).

### 3.3. RNF125 Is a Bona Fide Tumor Suppressor in Mouse and Human HCC

To further study the role of RNF125 in HCC, we tested whether RNF125 knockdown confers tumorigenic potential to non-tumorigenic cells. To this end, we used immortalized mouse liver progenitor cells (LPCs) established by the Lowe lab [10]. Lentiviral transduction of LPC cells with RNF125 shRNAs significantly downregulated RNF125 expression (Appendix A). Two weeks after subcutaneous injection of 1.0 × 10^6^ RNF125-depleted LPCs into the flanks of mice, visible tumors started to form in transplanted mice, while LPCs transduced with negative control (NC) shRNA did not induce tumors (Figure 2A), clearly indicating that RNF125 acts as a tumor suppressor. Next, we wanted to verify this tumor suppressive function in human liver. However, there were no non-transformed human liver cell lines available for analysis. We thus decided to use commercially available human liver cancer cell lines for these studies. None of the liver cancer cell lines used in these studies had mutations in RNF125 [39]. We chose 2 cell lines, HepG2 and SNU-398, based on the relatively high expression levels of RNF125 and HBV status (negative vs. positive). We first confirmed that lentiviral infection of RNF125 shRNAs significantly downregulated RNF125 expression in both cell lines (Appendix A). Although both cell lines are tumorigenic in transplanted mice, shRNA knockdown of RNF125 significantly accelerated their tumor growth in transplanted mice (Figure 2B,C). Collectively, these results demonstrated that RNF125 is as a tumor suppressor in mouse and human livers, regardless of the viral infection status, which is consistent with our transposon screening results. We next tested whether overexpression of RNF125 exerts a tumor suppressive function in human cancer cell lines. Transfection of a lentiviral RNF125 expression plasmid upregulated RNF125 in Hep3B cells and significantly reduced their tumor growth in vivo (Appendix A and Figure 2D). Similarly, but intriguingly, PLC/PRF/5 cells completely lost their tumorigenic capacity in vivo upon RNF125 overexpression (Appendix A and Figure 2E). These results suggested that introduction of RNF125 may have a therapeutic potential in human HCC.

### 3.4. RNF125 Negatively Regulates Cell Proliferation In Vitro

We next examined the mechanism of tumor suppression by RNF125. Tenorio et al. discovered RNF125 deletions and missense mutations in patients with overgrowth syndrome, in which most parameters of growth and physical development are above the means for age and sex [40]. This led us to check the functional involvement of RNF125 in cellular proliferation. Knockdown of RNF125 expression significantly accelerated the proliferation of HepG2 and SNU-398 cells, as exhibited by the WST-1 assay in vitro (Figure 3A,B). In contrast, overexpression of RNF125 significantly reduced the proliferation of Hep3B and PLC/PRF/5 cells (Figure 3C,D). These findings demonstrated that RNF125 negatively regulates cell proliferation in vitro.

### 3.5. RNF125 Transcriptionally Suppresses the Global Cell Proliferation Machinery

We next asked how RNF125 expression suppresses cell proliferation. Although RNF125 is known to function as a E3 ubiquitin ligase [41], there are multiple zinc finger domains in this protein as well [42], suggesting that it might also function as a transcription factor. Indeed, whole-transcriptome analysis of HepG2 cells showed that 33 genes were upregulated 2-fold or more by RNF125 knockdown (Figure 4A, Appendix A). These genes included Met, Egfr and Il6r, which are tyrosine kinase receptors important for liver regeneration [43]. We further confirmed that RNF125 knockdown upregulated Egfr and Met expression at the protein as well as mRNA level (Figure 4B–D). In addition, using Ingenuity Pathway Analysis (IPA), we analyzed the biological functions of 33 upregulated genes. Intriguingly, genes linked to cell proliferation were significantly enriched in this analysis (23 out of 33 genes, *p* < 3.79 × 10^−8^) (Figure 4E), suggesting global transcriptional regulation of cell proliferative genes by RNF125. 

### 3.6. RNF125 Negatively Regulates Cell Proliferation through Inhibition of the MET and EGFR Pathways

To examine the role of the MET and EGFR pathways in RNF125-mediated proliferation of HepG2 cells, we blocked the Met pathway using the small molecule Met inhibitor, Tivantinib, and observed little effect on proliferation of HepG2 cells (Figure 5A). In contrast, the increased proliferation of HepG2 cells observed upon RNF125 depletion was partially suppressed by Tivantinib treatment (Figure 5A), confirming that RNF125 negatively regulates cell proliferation in part through the inhibition of the Met pathway. Similar effects were also observed when Met shRNA was substituted for Tivantinib, confirming that the effects observed with Tivantinib were mediated by inhibition of the Met pathway (Figure 5B,C). Next, we blocked the EGFR pathway using the small molecule EGFR inhibitor Gefitinib. Unlike what we observed with Tivantinib, treatment of HepG2 cells with Gefitinib resulted in decreased proliferation of HepG2 cells (Figure 5D). However, unlike what we observed with Tivantinib, treatment with Gefitinib, significantly mitigated the increased growth of HepG2 cells observed in RNF125-depleted cells (Figure 5D). Likewise, dual siRNA knockdown of RNF125 and EGFR (Figure 5E) significantly alleviated the positive proliferative effects observed following RNF125 knockdown (Figure 5F). These data clearly show that RNF125 negatively regulates cell proliferation though the inhibition of the MET and EGFR pathways.

## 4. Discussion

Here we show that RNF125, a candidate cancer gene identified in liver-specific transposon screens performed in mice that carried or lacked a sensitizing HBs-Ag transgene, is a potent tumor suppressor gene in mouse and human HCC and functions as a negative regulator of cell proliferation. This gene, also known as TRAC-1, was originally discovered in a retroviral vector-based T cell surface activation marker screen [44]. A later study revealed that RNF125 functions as a RING finger ubiquitin E3 ligase in lymphoid tissue, and that its E3 ligase function is responsible for its positive regulation of T cell activation [45]. Arimoto et al. subsequently showed that RNF125 conjugates ubiquitin to the retinoic acid-inducible gene I (RIG-I), which facilitates its proteasomal degradation, indicating that RNF125 negatively regulates the antiviral immune response [41]. Another group demonstrated that RNF125 negatively regulates the replication of human immunodeficiency virus 1 (HIV-1) in primary human peripheral blood mononuclear cells [46]. Furthermore, RNF125 negatively regulates NLRP3 inflammasome and TRIM14-mediated innate immune response [47,48]. Although these studies revealed the importance of RNF125’s E3 ubiquitin ligase function in the field of immunology, the role of RNF125 in immune regulatory networks seems to be complex.

Kim et al. performed a loss-of-function screen to identify genes responsible for BRAF inhibitor (BRAFi) resistance in melanoma using siRNAs targeting 1173 ubiquitin proteasome system-related genes [49]. RNF125 was identified in this screen in combination with its downregulation in the BRAFi-resistant cell line. Using the BRAFi-resistant cell line, they found increased cell growth upon RNF125 depletion in the presence of BRAFi, but decreased cell growth with RNF125-overexpression, demonstrating that RNF125 is involved in the development of BRAFi-resistance in melanoma. Meanwhile, in this study, we found that cell proliferation was increased and decreased in HCC cell lines by RNF125 knockdown or overexpression, respectively, without any drug treatment, indicating that RNF125 is a bona fide negative regulator of cell proliferation, at least in the liver. Consistent with our findings, even in Kim’s study, RNF125 knockdown seemed to increase cell growth under low levels of BRAFi treatment (10^−10^ to 10^−9^ M), while RNF125 overexpression decreased growth in the UACC1113 melanoma cell line. In addition, Tenorio et al. recently identified deletion and missense mutations in RNF125 in OGS patients [40]. All missense mutations were predicted to be pathogenic or damaging mutations, and subsequent gene expression analysis revealed that RNF125 mRNA levels were low in these patients. Taken together, these data suggest that RNF125 may negatively regulate cell growth in a wide variety of tissue types, besides liver.

We also showed that RNF125 negatively regulates three tyrosine kinase receptors, MET, EGF receptor and the IL-6 receptor. HGF/Met, EGF and IL-6 pathways are known to be the most important pathways to trigger and accelerate liver regeneration [43], which is represented by the proliferation of quiescent hepatocytes. In the early phase of liver regeneration after experimental partial hepatectomy, IL-6 produced from Kupffer cells stimulates the IL-6/Stat3 pathway in the remaining hepatocytes and activates immediate-early genes such as Cyclin D1, Cyclin D3, c-Fos and c-myc. After this priming phase of liver regeneration, the HGF and EGF pathways are subsequently activated by their ligands and further induce cell growth and proliferation of primed hepatocytes through the Ras/Raf/Mek/Erk or Akt/mTOR pathways. Because these pathways are essential for hepatocytes to proliferate, it is not surprising to observe aberrant activation of these pathways in HCC and to consider them as potential therapeutic targets of HCC [50]. In this study, we performed in vitro experiments and showed that knockdown of either EGFR or MET alone significantly alleviated the increase in cell proliferation observed following RNF125 depletion. These results demonstrated that these pathways were indeed responsible for RNF125-meditated negative regulation of HCC cell growth. In addition, they also suggested the crosstalk between the EGF and HGF/Met pathways in the proliferation of malignant hepatocytes. It has been reported that the HGF/Met and EGF pathways share common downstream signaling pathways, including the PI3K/Akt and MAPK pathways [51]. In addition, Jo et al. reported that MET and EGFR were co-immunoprecipitated in tumor cells but not in normal hepatocytes. They also showed that EGFR ligand TGFα induced MET activation and inhibition of EGFR by neutralizing antibody or EGFR-specific inhibitor suppresses the MET pathway as well as the EGF pathway [52]. Collectively, these findings suggested crosstalk between these pathways in HCC, and that RNF125 is an important negative regulator of both pathways. In addition to MET and EGFR, we also found that multiple genes promoting cell proliferation were also upregulated by RNF125 knockdown, including IL6R, MCL1, AKR1B10 and SOX4 [53,54,55]. These findings suggested that RNF125 may act as a master regulator of cell proliferation in HCC cells.

It will be interesting to understand how RNF125 regulates these pathways transcriptionally. Kim et al. found that RNF125 knockdown upregulates EGFR expression in melanoma cell lines [49]. However, EGFR is not a direct substrate of RNF125, rather it seems to be transcriptionally and post-translationally regulated by RNF125. Kim also showed that JAK1 is a substrate of RNF125’s ubiquitin ligase activity and is stabilized in RNF125-depleted cells, which is partially responsible for EGFR’s post-translational upregulation. In our study, in addition to known molecules crucial for liver regeneration, including EGFR, transcriptome analysis showed that RNF125 negatively regulates multiple molecules important for cell proliferation, suggesting the existence of global transcriptional regulation of proliferative genes. One possible explanation for this phenomenon is that RNF125 ubiquitinates and degrades an important transcriptional factor (TF) involved in cell proliferation. To pursue this possibility, using the public analysis tool “MotifMap” [56], we comprehensively searched the TF binding sites around the promoter regions of the 33 genes that were transcriptionally upregulated in RNF125-depleted cells (Appendix A). We also used the database TFactS [57] to predict upstream TF binding sites from the list of differentially expressed genes (Appendix A). Unfortunately, we did not find any good candidate TFs that were consistently identified in both analyses. We also tried to search the gene candidates between RNF125 and multiple proliferative genes using the IPA platform. Regulator effect analysis predicted that IL1B, ERK and HIF1A might be intermediates that account for the global upregulation of genes involved in cell proliferation. Indeed, knockdown of RNF125 significantly upregulated the expression levels of IL6R and phosphorylated ERK (Appendix A), suggesting the involvement of these molecules in RNF125-mediated global regulation of proliferative genes. However, it is unclear how RNF125 might regulate these genes. Another possibility is that RNF125 itself functions as a novel transcriptional factor. Indeed, this possibility is raised by the fact that the C-terminal region of RNF125 contains two basic C2H2 zinc-finger motifs, although these motifs are not classical DNA-binding zinc fingers [42]. Further investigation will therefore be needed to clarify this regulatory network.

Two sets of hHuman data clearly showed that RNF125 expression levels are significantly downregulated in HCC (Figure 1B,D). However, the mechanism behind this downregulation is unknown. In general, gene inactivation in cancer is caused by somatic mutations, copy number aberrations, and epigenetic changes. Examination of the somatic mutation data for 503 HCCs, revealed that there was only one non-silent mutation in RNF125 [7]. TCGA copy number analysis also identified 33 significant focal deletions among 370 HCCs, however none of these regions contains the ZNF125 locus [58]. Villanueva et al. performed methylome profiling of 304 HCCs but didn’t find any hypermethylated regions within the promoter region of RNF125 [59]. We also searched for possible miRNA regulation of RNF125. We first looked for significantly upregulated miRNAs in HCC. We analyzed three publicly available miRNA microarray human HCC datasets (GSE6857, GSE10694 and GSE22058) and found 22 miRNAs showing more than 1.2-fold upregulation. Among them, 4 miRNAs were predicted to target RNF125 using three miRNA target prediction programs (TargetScan, miRanda and microT-CDS) (Appendix A). miR-15b-5p is the only miRNA that was upregulated in all 3 microarray datasets and by all three prediction programs. Zhu et al. reported that miR-15b-5p is upregulated following Japanese Encephalitis Virus (JEV) infection and induces RIG-I expression through negative regulation of RNF125 [60]. These results clearly demonstrate that miR-15b-5p directly binds to the RNF125 3’UTR and downregulates RNF125 transcription. They also suggest that upregulation of miR-15b-5p may be at least partially responsible for RNF125 downregulation in HCC.

In this study, we have several limitations. First, regarding the functional role of RNF125 as a tumor suppressor, we have only focused on cell proliferation. Hepatocarcinogenesis requires a variety of process including apoptosis resistance, anoikisis evasion, migration, invasion, vascular formation, immune evasion, metabolic adaptation, and so on [61]. Further studies are desired to reveal the involvement of RNF125 in these processes. Second, although we have confirmed the clinical relevance of RNF125 in 2 publicly available datasets of human HCC patients, we did not check it in our own dataset. Because we have only access to the limited information in the publicly available datasets, we thus cannot provide the relationship between RNF125 and detailed clinical characteristics.

## 5. Conclusions

In summary, we identified and validated a new anti-proliferative tumor suppressor gene, RNF125, using transposon mutagenesis screens, comparative genomics, and functional studies. RNF125 is a negative regulator of multiple genes important for cell proliferation and liver regeneration. Expression of RNF125 is frequently downregulated in HCC, and its inactivation has a significant negative impact on patient prognosis. Based on these findings, we propose a new cell proliferation regulatory network mediated by RNF125 that contributes to HCC development and progression.

## Figures and Tables

**Figure 1 cancers-14-02589-f001:**
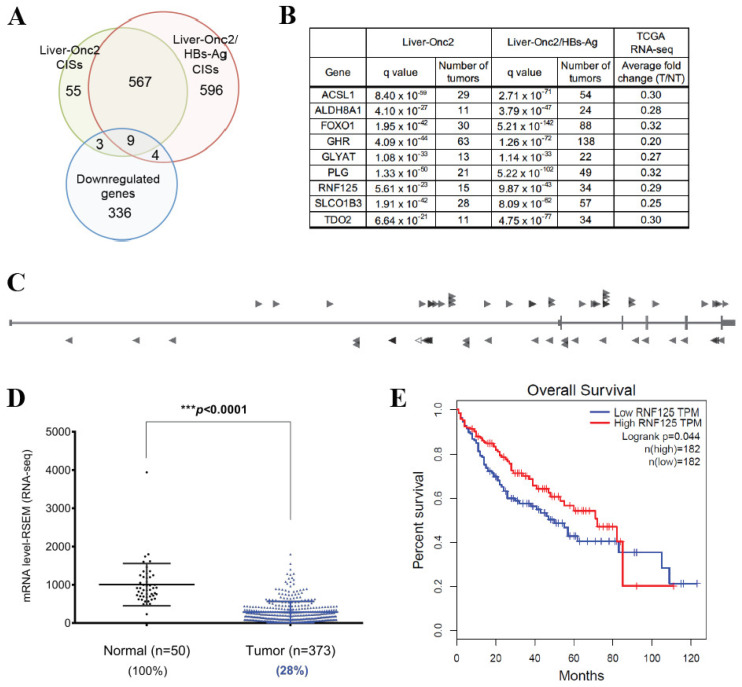
RNF125 is a candidate tumor suppressor and significantly downregulated in human HCC. (**A**) Cross-species comparison among stringent CISs of tumors from Alb-Cre/+; T2Onc2/+; Rosa26-lsl-SB11/+; HBs-Ag/+ mice (Liver-Onc2/HBsAg mice) and those from Liver-Onc2 mice, and genes with more than 3-fold downregulation of mRNA abundance in human HCC. (**B**) Nine common genes. (**C**) Pattern of Sleeping Beauty transposon insertions indicated that RNF125 is a tumor suppressor. (**D**) RNF125 is significantly downregulated in human HCC. (**E**) Low expression levels of RNF125 are associated with poor patient survival in HCC.

**Figure 2 cancers-14-02589-f002:**
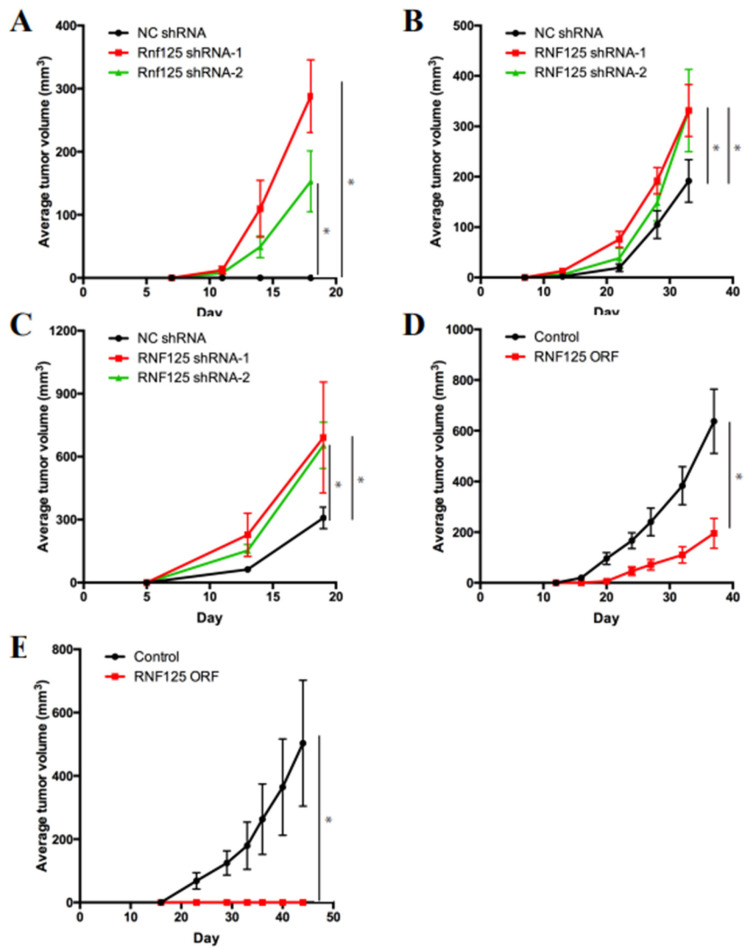
RNF125 inhibits HCC tumor development. (**A**) Knockdown of RNF125 in mouse immortalized liver progenitor cells (LPCs) led to accelerated xenograft tumor growth in athymic nude mice (* *p* < 0.05). (**B**) Knockdown of RNF125 in HepG2 HCC cells led to accelerated xenograft tumor growth in athymic nude mice (* *p* < 0.05). (**C**) Knockdown of RNF125 in SNU-398 HCC cells led to accelerated xenograft tumor growth in athymic nude mice (* *p* < 0.05). (**D**) Overexpression of RNF125 in PLC/PRF/5 HCC cells led to decelerated xenograft tumor growth in athymic nude mice (* *p* < 0.05). (**E**) Overexpression of RNF125 in HepB3 HCC cells led to decelerated xenograft tumor growth in athymic nude mice (* *p* < 0.05).

**Figure 3 cancers-14-02589-f003:**
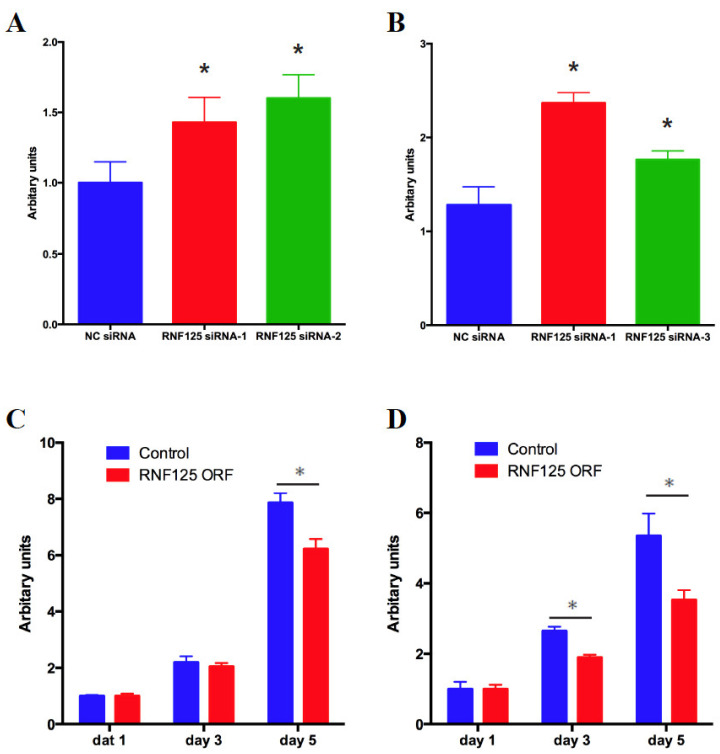
RNF125 inhibits HCC cell proliferation. HepG2 cells (**A**) and SNU-398 cells (**B**) were transfected with negative control (NC) siRNA or RNF125 siRNAs. WST-1 assay was used to measure the proliferation of these cells 4 days after transfection (* *p* < 0.05). PLC/PRF/5 cells (**C**) and Hep3B cells (**D**) were lentivirally transduced with control vector or RNF125 expression vector. Cell proliferation was assessed by WST-1 assay (* *p* < 0.05).

**Figure 4 cancers-14-02589-f004:**
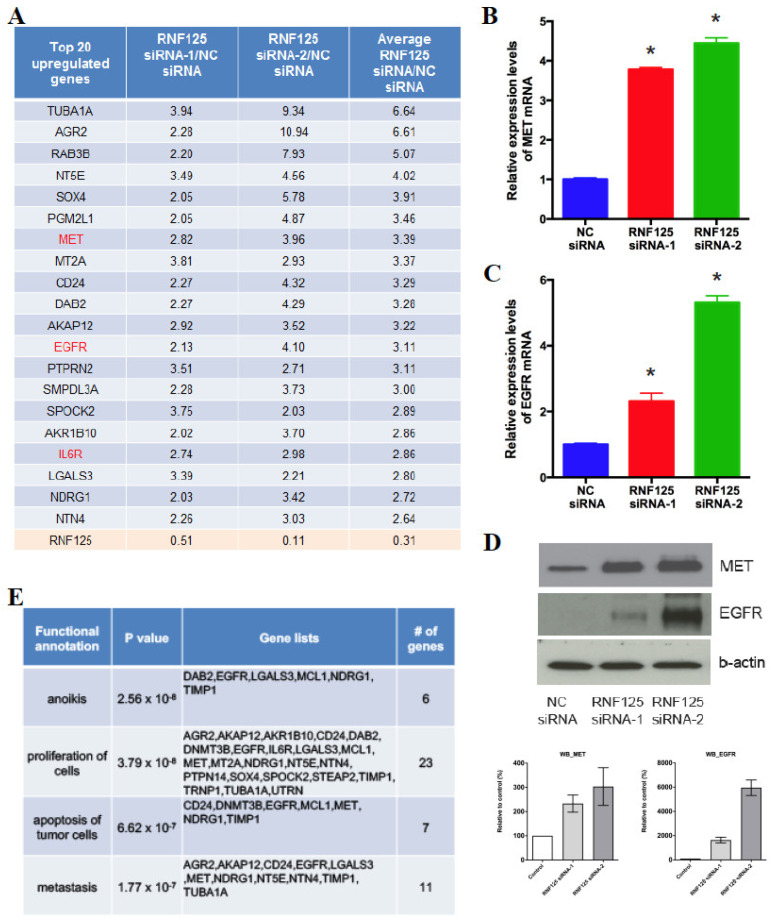
RNF125 transcriptionally suppresses global cell proliferation machinery. (**A**) HepG2 cells were transfected with negative control (NC) siRNA or RNF125 siRNA. mRNA abundance was assessed by ribosomal-depleted RNA-seq 3 days after transfection. List of 23 genes showed >2.5-fold upregulation of mRNA abundance in average expression levels between RNF125 siRNA-1 and siRNA-2 compared to NC siRNA (**A**). (**B**–**D**) HepG2 cells were transfected with NC siRNA or RNF125 siRNA and cultured for 2 days. Met (**B**) and Egfr (**C**) mRNA expression levels was assessed by qPCR (N = 4, * *p* < 0.05 vs. NC siRNA). Egfr and Met protein expression levels was assessed by western blot (**D**). (**E**) HepG2 cells were transfected with NC siRNA or RNF125 siRNA. mRNA abundance was assessed by ribosomal-depleted RNA-seq 3 days after transfection. The Ingenuity Ontology analysis of 33 genes with >2-fold upregulation of mRNA abundance in relative expression levels of both RNF125 siRNA-1 and siRNA-2 compared to NC siRNA. Enriched biological functions were listed with *p*-value.

**Figure 5 cancers-14-02589-f005:**
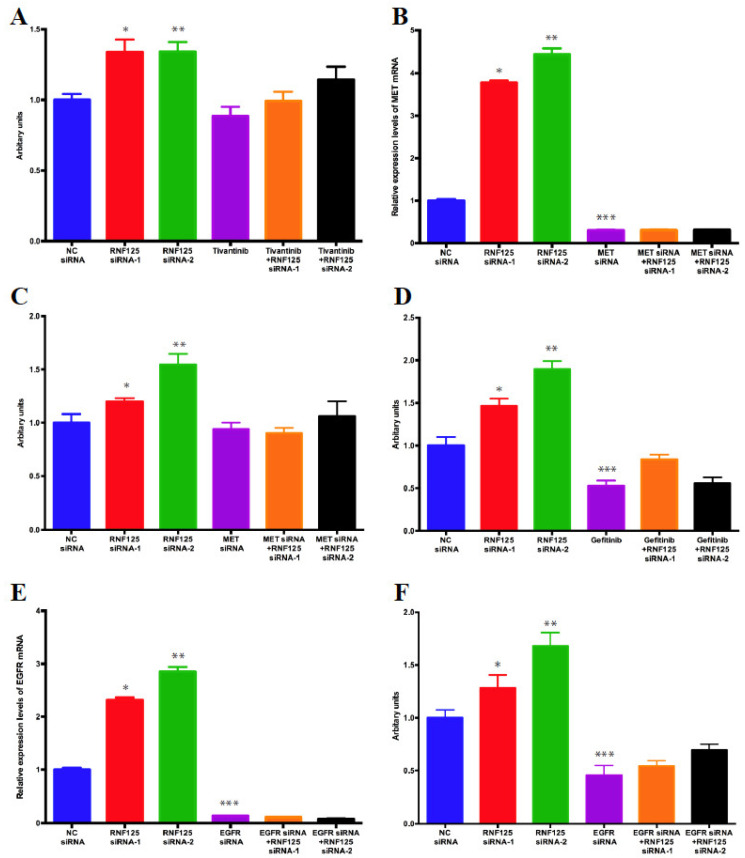
RNF125 suppresses cell proliferation through inhibiting the Egfr and Met pathway. (**A**) HepG2 cells were transfected with negative control (NC) siRNA or RNF125 siRNA. One day after transfection, 5 uM of tivantinib or DMSO was added to the well and incubated for 3 more days. Cell proliferation was assessed by WST-1 assay (N = 4, * *p* < 0.05 vs. NC siRNA and Tivantinib + RNF125 siRNA-1, ** *p* < 0.05 vs. NC siRNA and Tivantinib + RNF125 siRNA-2). (**B**,**C**) HepG2 cells were transfected with NC siRNA or RNF125 siRNA and/or Met siRNA. mRNA abundance of Met gene was assessed by qPCR 2 days after transfection (N = 4, * *p* < 0.05 vs. NC siRNA and MET siRNA + RNF125 siRNA-1, ** *p* < 0.05 vs. NC siRNA and MET siRNA + RNF125 siRNA-2, *** *p* < 0.05 vs. NC siRNA) (**B**). Cell proliferation was assessed by WST-1 assay 4 days after transfection (N = 4, * *p* < 0.05 vs. NC siRNA and MET siRNA + RNF125 siRNA-1, ** *p* < 0.05 vs. NC siRNA and MET siRNA + RNF125 siRNA-2) (**C**). (**D**) HepG2 cells were transfected with NC siRNA or RNF125 siRNA. One day after transfection, 10 uM of erlotinib or DMSO was added to the well and incubated for 3 more days. Cell proliferation was assessed by WST-1 assay (N = 4, * *p* < 0.05 vs. NC siRNA and Gefitinib + RNF125 siRNA-1, ** *p* < 0.05 vs. NC siRNA and Gefitinib + RNF125 siRNA-2, *** *p* < 0.05 vs. NC siRNA). (**E**,**F**) HepG2 cells were transfected with NC siRNA or RNF125 siRNA and/or EGFR siRNA. mRNA abundance of Egfr gene was assessed by qPCR 2 days after transfection (N = 4, * *p* < 0.05 vs. NC siRNA and EGFR siRNA + RNF125 siRNA-1, ** *p* < 0.05 vs. NC siRNA and EGFR siRNA + RNF125 siRNA-2, *** *p* < 0.05 vs. NC siRNA) (**E**). Cell proliferation was assessed by WST-1 assay 4 days after transfection (N = 4, * *p* < 0.05 vs. NC siRNA and EGFR siRNA + RNF125 siRNA-1, ** *p* < 0.05 vs. NC siRNA and EGFR siRNA + RNF125 siRNA-2, *** *p* < 0.05 vs. NC siRNA) (**F**).

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
