# Peer review of "Ring Finger Protein 125 Is an Anti-Proliferative Tumor Suppressor in Hepatocellular Carcinoma"

_cancers, 2022, doi:10.3390/cancers14112589_

Round 1

Reviewer 1 Report

The authors previously performed a SB transposon mutagenesis screening and revealed a series of HCC candidate cancer genes based on a HBV-HCC mouse model. In this study, they further dig the current HCC databases from TCGA, GEO and other platform and found a new TSG, rnf125. The in vitro experiment showed that rnf125 could inhibit HCC proliferation possibly via Egfr and Met. The manuscript was well orgnized and clearly written. However,  there were some shortcomings that limited the value of this study.  

  1. The author did not provide sufficient evidence for choosing rnf125 for further investigation.  "Rnf125 was linked to neither HCC nor metabolism" seemed to be not a good reason.  They need to select proper candidate gene according to the statistical significance and clinical value. In TCGA databases, dozens of genes could do better than rnf125 in prognostic stratification of HCC patients. 
  2. The functional exploration of rnf125 was also insufficient. The author need to correlated rnf125 with tumor features. Is there any postive relationship? How about the role of rnf125 in invasiveness, metastasis, etc? In vivo experiment is indispensable.
  3.   The downstream pathways that transcriptional regulated by rnf125 have been showed in several previous studies. There are far more than Met and Egfr. Why only choose the two? Did they performed a transcriptome using in vitro model?
  4. The whole story including the expression and clinical value of rnf125 and its downstream molecules  should be verified in their own cohorts particularly HBV-HCC patients.

Author Response

Please see the attachment for detailed response to your comments. Thanks!

Reviewer 2 Report

The authors identified and validated a new anti-proliferative 527 tumor suppressor gene, RNF125, using transposon mutagenesis screens, 528 comparative genomics, and functional studies. RNF125 is a negative 529 regulator of multiple genes important for cell proliferation and liver 530 regeneration. Although I have no practical experience of transposon mutagenesis screens, the studies seem to have been competently conducted. This manuscript should be published when substantiated by better arguments.

The main purpose of this study is to identify HCC driver genes and they performed transposon mutagenesis screens in a mouse HBV model of HCC and discovered many candidate cancer genes. I consider the topic is relevant in the field because it is difficult to identify driver genes for HCC under recent advances in sequencing technology . The important aspects of this study are to identify and validate a new anti-proliferative tumor suppressor gene, RNF125, using transposon mutagenesis screens, comparative genomics, and functional studies, which are specific improvements could the authors consider regarding the methodology. The conclusions consistent with the evidence and arguments presented and they address the main question posed. If I may say so, the text in Figure 5 has poor resolution and is difficult to read, and I would like to see it corrected.

Author Response

(The authors gave the same response as above.)

Round 2

Reviewer 1 Report

The authors nicely addressed all the questions. Congratulations. It is OK for me now.